# Inversion of Shear and Longitudinal Acoustic Wave Propagation Parameters in Sea Ice Using SE-ResNet

**DOI:** 10.3390/s25185663

**Published:** 2025-09-11

**Authors:** Jin Bai, Yi Liu, Xuegang Zhang, Wenmao Yin, Ziye Deng

**Affiliations:** 1Science and Technology on Underwater Test Control Laboratory, Dalian 116000, China; 15524433577@163.com (J.B.); yinwenmao760@sina.com (W.Y.); 2School of Marine Science and Technology, Tianjin University, Tianjin 300072, China; 2023227035@tju.edu.cn (Y.L.); 15524626441@163.com (Z.D.)

**Keywords:** sea ice, acoustic parameter measurement, ResNet

## Abstract

With the advancement of scientific research, understanding the physical parameters governing acoustic wave propagation in sea ice has become increasingly important. Among these parameters, shear wave velocity plays a crucial role. However, as measurements progressed, it became apparent that there was a large discrepancy between measured values of shear waves and predictions based on empirical formulas or existing models. These inconsistencies stem primarily from the complex internal structure of natural sea ice, which significantly influences its physical behavior. Research reveals that shear wave velocity is not only influenced by bulk properties such as density, temperature, and stress state but is also sensitive to microstructural features, including air bubbles, inclusions, and ice crystal orientation. Compared to longitudinal wave velocity, the characterization of shear wave velocity is far more challenging due to these inherent complexities, underscoring the need for more precise measurement and modeling techniques. To address the challenges posed by the complex internal structure of natural sea ice and improve prediction accuracy, this study introduces a novel, integrated approach combining simulation, measurement, and inversion intelligent learning model. First, a laboratory-based method for generating sea ice layers under controlled formation conditions is developed. The produced sea ice layers align closely with measured values for Poisson’s ratio, multi-year sea ice density, and uniaxial compression modulus, particularly in the high-temperature range. Second, enhancements to shear wave velocity measurement equipment have been implemented. The improved device achieves measurement accuracy exceeding 1%, offers portability, and meets the demands of high-precision experiments conducted in harsh polar environments. Finally, according to the characteristics of small sample data. The ANN neural network was improved to a deep residual neural network with the addition of Squeeze-and-Excitation Attention (SE-ResNet) to predict longitudinal and transverse wave velocities. This prediction method improves the accuracy of shear and longitudinal wave velocity prediction by 24.87% and 39.59%, respectively, compared to the ANN neural network.

## 1. Introduction

The polar regions, rich in resources such as oil, natural gas, and rare metals, have become critical focal points for scientific research in the 21st century [1]. At the same time, these regions are undergoing unprecedented environmental changes due to global warming, marked by the pronounced “polar amplification effect” [2]. Understanding the physical and acoustic properties of sea ice is essential, as sea ice plays a fundamental role in climate regulation, resource exploration, and environmental monitoring.

Sea ice serves as a medium for acoustic wave propagation, which is critical for applications such as underwater communication, sonar systems, and polar exploration. Among its acoustic parameters, shear wave velocity is particularly significant, as it directly impacts the accuracy of acoustic wave propagation modeling in sea ice. However, current predictive models often deviate significantly from experimental measurements, limiting their reliability. For example, Lee et al. reported discrepancies in predictions of sea ice mass balance and sea-level changes due to limitations in empirical models [3], while Meng et al. demonstrated the randomness and limitations of empirical formulas in predicting the compressive strength of sea ice [4]. These inconsistencies underscore the need for more advanced methods to accurately model and invert the physical parameters of sea ice.

The primary challenge lies in the complex internal structure of natural polar sea ice, which evolves through intricate thermodynamic processes and significantly influences its mechanical and acoustic properties [5,6]. Factors such as snow accumulation, air bubbles, inclusions, and ice crystal orientation add to the complexity of shear wave velocity measurement and prediction [7]. Compared to longitudinal wave velocity, shear wave velocity is far more challenging to characterize due to these microstructural features and harsh polar environmental conditions [8,9,10].

Recent advances in intelligent learning models provide powerful tools for solving such complex regression problems. ANN neural networks are undoubtedly the best choice for a form of parameter-to-result prediction, such as intra-ice sound speed prediction, where there is an intrinsic physical mechanism link. Good reviews of ANN modeling in hydrology were presented at the ASCE 33rd Task Committee on Applications of Artificial Neural Networks in Hydrology [11] and Dawson and Wilby 34th [12]. However, for small sample data, a slight increase in the number of ANN neural network training alone will result in overfitting. Therefore, there is a need to improve ANN neural networks to form deep neural networks, which have been remarkably successful in modeling nonlinear relationships and managing complex parameter interactions in a variety of fields, including polar research [13,14,15]. Intelligent learning models such as deep neural networks offer a promising alternative to cope with parameter prediction with higher accuracy and computational efficiency.

To overcome these limitations, this study proposes an intelligent learning-based framework that integrates laboratory experiments, advanced measurement techniques, and deep learning methods for the simulation and inversion of shear and longitudinal wave propagation parameters in polar sea ice. The main contributions of this study are as follows:**Laboratory Sea Ice Preparation**: A method for generating sea ice layers under controlled thermodynamic conditions is developed, ensuring consistency with real-world properties such as density, Poisson’s ratio, and mechanical strength.**Measurement Accuracy Enhancement**: A portable and highly accurate shear wave velocity measurement device is introduced, improving precision and addressing practical challenges in polar-like environments.**Deep Residual Neural Network with the Addition of Squeeze-and-Excitation Attention (SE-ResNet)**: Improvement of the initial ANN neural network using the ResNet network, which can increase the depth of the network infinitely to cope with the needs of small samples and deeper learning. Also, we add Squeeze-and-Excitation Attention. Compared with traditional ANN network models, the intelligent learning framework improves prediction accuracy by 24.87% and 39.59%.

By utilizing ResNet and SE Attention intelligent learning techniques, this study not only advances the accuracy of shear wave velocity prediction but also provides a scalable and efficient solution for simulating acoustic wave propagation in polar sea ice. The proposed approach lays a robust foundation for polar acoustic research and contributes to broader applications, including underwater exploration, sonar detection, and environmental monitoring in extreme environments.

## 2. Materials and Methods

### 2.1. Laboratory Preparation of Sea Ice Samples

The complex structure of natural sea ice, which evolves through thermodynamic processes, significantly influences its acoustic properties, particularly in the propagation of shear waves [16,17]. Natural sea ice can be classified into granular or columnar ice based on its growth patterns, and its salinity distribution varies with age and depth. For instance, one-year sea ice typically exhibits higher salinity, with the surface salinity (3.2–7.2 ppt) being considerably lower due to salt expulsion. The surface layer accounts for 25 ± 12% of the total ice thickness. In contrast, multi-year sea ice has much lower salinity (0.4–2.4 ppt), with the freshwater surface layer occupying 30 ± 9% of the total thickness [18,19]. These characteristics make the preparation of laboratory sea ice samples that mimic natural conditions critical for studying sea ice acoustic behavior.

Challenges in Conventional Sea Ice Preparation

In traditional freezers, heat exchange during ice formation is uncontrolled, leading to the formation of gas bubbles and uneven salt distribution in the sea ice (Figure 1b). This differs significantly from the natural freezing process, where heat is exchanged predominantly from the surface, and salt precipitates gradually into seawater (Figure 1a). As a result, conventional ice-making methods fail to replicate the stratified salinity distribution observed in natural sea ice.

Improved Sea Ice Preparation Method

To simulate polar sea ice more accurately, this study developed an improved sea ice preparation container (Science and Technology on Underwater Test Control Laboratory, Dalian, China) to control heat exchange directionally. The apparatus ensures that heat transfer occurs only from the surface, replicating the natural sea ice formation environment.

The improved container (Figure 1c) consists of a thermostatic foam protective layer and a thickness measurement device. Brine solutions, labeled ICE1 and ICE2, were prepared based on different salinity ratios (6.5 ppt and adjusted values) to achieve realistic salt precipitation characteristics. The freezing process follows these steps:Temperature Control: The cryogenic laboratory temperature is initially set to 0 °C to stabilize the liquid, after which it is progressively lowered to simulate polar conditions.Salt Precipitation: During freezing, 70% to 90% of the salt precipitates, depending on temperature. The salt precipitation can be expressed as follows:(1)$$Q=k\left(T+1.8\right)S$$
where Q is the amount of precipitated salt, k is the proportional constant, and T is the freezing temperature.

3.Layered Ice Simulation: The freezing process ensures stratification, creating sea ice layers with different salinity distributions. The measured salinity profiles closely resemble those observed in natural one-year and multi-year sea ice.

Upon completion, the prepared sea ice layers were cut and segmented for analysis, ensuring consistency in thickness and salinity profiles. Figure 2a shows the thickness measurements of ice made from different solutions, and Figure 2b shows the ice split layering.

Simulated Sea Ice Types

In this study, laboratory-simulated ice was classified into four types based on the observed general stabilization temperatures and salinities of different types of sea ice [20,21,22], as summarized in Table 1. The categorization here is intended only as a simple classification condition for the different types of ice that can be made in laboratory ice production. The type of cycle is determined primarily based on the temperature of the sea ice growth and decay cycle. The year of the sea ice is determined by the difference in salinity between one-year and multi-year ice.

These simulated types cover the full lifecycle of sea ice, including decaying, one-year, and multi-year phases, providing a robust foundation for acoustic parameter measurements under realistic conditions.

By improving the laboratory preparation process, the method is closer to the physical and structural properties of natural sea ice, allowing the resulting sea ice to model acoustic parameters more accurately.

### 2.2. Measurement of Acoustic Parameters of Sea Ice Layer

Accurate measurement of acoustic parameters, such as shear wave velocity and longitudinal wave velocity, is essential for understanding acoustic wave propagation in sea ice and improving the precision of related models. However, traditional measurement techniques face limitations: natural sea ice conditions are difficult to replicate in laboratories, and existing equipment often fails to provide reliable data under complex polar conditions. To address these challenges, we developed and optimized an experimental setup that enables controlled, precise measurements of acoustic parameters in laboratory-prepared sea ice layers.

#### 2.2.1. Development of Acoustic Parameter Measurement Equipment

The measurement system was designed to quantify the shear wave velocity and longitudinal wave velocity of sea ice under controlled conditions, optimizing the measurements of Kim and Vogt et al. [23,24,25]. The system consists of four main components (Figure 3).

Acoustic Wave Transmission Unit (Xiangtan Electronic Research Institute, Hunan, China)

Generates sinusoidal signals using a 12-bit DAC with an 8 MHz sampling rate, capable of real-time waveform output.

Provides adjustable amplitude signals through a power amplification module housed in a standardized chassis with BNC connectors for signal input/output.

2.Acoustic Wave Reception Unit (Science and Technology on Underwater Test Control Laboratory, Dalian, China)

Captures and processes signals using a trigger module and an FPGA-controlled data acquisition system.

Filters and amplifies analog signals before converting them into digital signals via an A/D conversion module for further analysis.

3.Acoustic Transducer Unit (Science and Technology on Underwater Test Control Laboratory, Dalian, China)

Transmitting transducers with radial vibration modes at low frequencies and receiving transducers oriented parallel to each other to detect transverse wave delay.

Matching circuits ensure optimal transducer performance, improving signal accuracy.

4.Control Unit (Xiangtan Electronic Research Institute, Hunan, China)

Manages data transmission and system synchronization through a programmable control interface.

The experimental setup, illustrated in Figure 4a, ensures reliable signal transmission and reception while minimizing environmental interference. The transducers are fixed at both ends of the sea ice sample, as shown in Figure 4b, where the transducers are fixed to the ends of the long axis of the sea ice sample, and a coupling agent is applied to the surface to ensure safe and effective contact between the transducers and the sea ice surface.

Measurement of the shear wave speed of sound requires the use of special transducers, the main vibration mode for radial vibration, that is, most of the energy in the form of shear wave transfer, as shown in Figure 5a.

The acoustic parameter measurement set consists entirely of stand-alone components, which facilitates the conduct of polar in situ measurement experiments.

#### 2.2.2. Measurement of Acoustic Parameters

The system performs sound velocity measurements by measuring the difference between the take-off time of the acoustic signal and the signal emission time. The calculation equation is as follows.(2)$$V=\frac{\left|L-L_{0}\right|}{t-t_{0}}$$
where V is the speed of sound in the ice sample, L is the sample length, $L_{0}$ is the zero point of the scale, t is the take-off time, and $t_{0}$ is the transducer-corrected moment of 0. Depending on the vibration mode of the signal, the shear wave velocity $V_{s}$ and longitudinal wave velocity $V_{l}$ can be distinguished

The transmitted signal contains three sinusoidal signal cycles per 1 ms waveform, and each measurement is repeated 100 times to increase the signal-to-noise ratio of the pulses. The average and amplitude difference of each sample are determined, allowing the received signal to be smoothed without affecting the original signal. This method can effectively improve the measurement accuracy.

The results and measurement errors before and after repeated measurements are shown in Figure 6. The frequency used for this error estimation is 100 kHz. The take-off time error range is ±0.004 ms for a single measurement, the measured length error range of ±0.3 mm, and the error range is ±48 m/s. The curve is obviously smooth after averaging 100 measurements of the same sample, at which time the error range of the take-off time is ±0.001 ms, and the error range of the sound speed is within ±12 m/s. The actual measurement was performed with a sample length of 100 cm to accommodate signals of different frequencies.

In addition, the study conducted experiments on the key influencing factors: frequency and incident angle. An example of relevant measurement results is shown in Figure 7, where we have completed the integration and visualization of the measurement program (Science and Technology on Underwater Test Control Laboratory, Dalian, China)for the measurement device.

For frequency, due to the limitations of the fixed transducer frequency, the longitudinal sound velocity measurements were performed at four frequency points: 25 kHz, 50 kHz, 75 kHz, and 100 kHz. The shear sound velocity measurements were performed at two characteristic frequency points: 44 kHz and 50 kHz.

For the angle of incidence, the structural anisotropy of sea ice, caused by vertical growth and phase components, introduces directional variations in its acoustic properties [26]. We chose 0° incidence angle perpendicular to the sea ice growth direction and 180° incidence angle parallel to the sea ice growth direction for our study.

By systematically varying the frequency and angle of incidence, it is possible to determine the effect of internal heterogeneity in the sea ice on wave velocity.

### 2.3. SE-ResNet for Shear Wave Parameter Simulation and Inversion

In this study, we propose a deep residual neural network with the addition of Squeeze-and-Excitation Attention (SE-ResNet) for the prediction and inversion of sea ice acoustic propagation parameters, including shear wave and longitudinal wave velocities.

The SE-ResNet architecture is built by embedding compressed Squeeze-and-Excitation Attention and residual blocks into a multi-layer neural network to enable deeper learning while maintaining computational efficiency. The main components of the architecture are shown in Figure 8 and detailed below:

SE-ResNet neural network can be divided into four parts: data processing, ANN, SE Attention model, and ResNet model.

Data Processing

The data processing section normalizes the eight feature parameters of ice thickness, salinity, density, temperature, elastic modulus, Poisson’s ratio, acoustic frequency, and incident angle. Normalization is performed to ensure stable training and faster convergence. The normalized dataset was randomly split in the ratio of 8:2 for training and testing. The normalization is done in the following way.(3)$$D^{\prime }=\frac{D-min(D)}{\max\left(D\right)-min(D)}$$

2.ANN

The Adam optimizer was used to update ANN networks’ weights, with a learning rate scheduler dynamically adjusting the learning rate based on validation loss. Hyperparametric tuning of key parameters such as the number of neurons, dropout rate, and learning rate in ANN networks. A random search is conducted over 200 parameter combinations, and the optimal settings are determined as follows:

Fully connected layer neurons: 16

Dropout rate: 0.2

Hidden layer neurons: 112

Learning rate: 0.01

Minimizing the root-mean-square error (MSE) between predicted and actual values is used in the model evaluation

3.SE Attention Model

The SE Attention model consists of four main components, which are Global Average Pooling, Fully Connected Layer Group, Reshape layer, and Channel-wise Multiplication Layer.

The core of the SE Attention model is the channel reduction capability of its fully-connected layer and the adaptive assignment of weights to each feature channel of the Channel-wise Multiplication Layer.

The two fully connected layers are defined as follows:

The first fully connected layer reduces the number of channels by a factor of $R_{D}$ and its activation function is ReLU. The second fully connected layer reduces the number of channels, and its activation function is Sigmoid.(4)$$s_{1}=ReLU\left(W_{1}z+b_{1}\right),s_{1}\in R^{B\times \frac{C}{R_{D}}}$$(5)$$s_{2}=\sigma \left(W_{2}S_{1}+b_{2}\right),s_{2}\in R^{B\times C}$$
where σ represents the Sigmoid activation function, and $W_{1}$, $b_{1}$, $W_{2}$, $b_{2}$ is the weight and bias of the fully connected layer.

In Channel-wise Multiplication Layer, the adjusted feature weights are multiplied channel-by-channel with the inputs:(6)$$Y=X\cdot s_{2}$$

X denotes the input tensor, Y denotes the output tensor, and $s_{2}$ is the attention weight tensor obtained after the Squeeze-and-Excitation mechanism.

4.ResNet Model

The ResNet model uses the classical ResNet-18 structure, consisting of four layers of residual blocks with progressively decreasing number of feature channels and a final Global Average Pooling layer to progressively aggregate low-level features.

The core of the ResNet model is its residual block, consisting of two convolutional layers with jump connectivity that bypass the middle layer. The jump connections allow gradients to propagate directly through the network and help mitigate gradient vanishing.

A single residual block is defined as follows:

First Convolutional Layer: Applies a 1D convolution operation with a kernel size of 3 to extract local features. The output is batch-normalized and passed through a ReLU activation function.(7)$$X_{conv1}=Conv1D(A_{2},filters=C_{out},kernel=3,padding=‘same’)$$(8)$$X_{bn1}=BatchNorm\left(X_{conv1}\right)$$(9)$$X_{relu1}=ReLU\left(X_{bn1}\right)$$

Second Convolutional Layer: Further processes the features using another 1D convolution, followed by batch normalization.$$X_{conv2}=Conv1D(X_{conv1},filters=C_{out},kernel=3,padding=‘same’)$$(10)$$X_{bn2}=BatchNorm\left(X_{conv2}\right)$$

Shortcut Connection: If the input and output dimensions of the residual block differ, a 1D convolution adjusts the dimensions of the input. The shortcut connection adds the adjusted input to the output of the second convolutional layer:(11)$$S=\left\{\begin{array}{cc} A_{2}, & if{C_{in}=C}_{out}\\ Conv1D\left(A_{2},filters=C_{out},kernel=1,padding=‘same’\right), & if{C_{in}\neq C}_{out} \end{array}\right.$$(12)$$X_{out}=ReLU(X_{bn2}+S)$$

## 3. Results

### 3.1. Physical Parameters of Laboratory-Prepared Ice Layers

Sea ice exhibits a heterogeneous and porous crystalline structure, making its properties more complex than freshwater ice. During the freezing process, sea ice forms unique layered structures [27,28,29]. To evaluate the effectiveness of laboratory-prepared ice layers, their physical parameters were measured in layers, including temperature, salinity, density, elastic modulus, and Poisson’s ratio.

Temperature measurements are stabilized by burying a probe in a hole in the ice; salinity measurements are measured by analyzing melted samples using a salinometer; density measurements are calculated from the volume and mass of standardized samples; Young’s modulus and Poisson’s ratio were measured by uniaxial compression experiments in transverse and longitudinal directions.

Through the above measurement methods, we can obtain the distribution of basic physical parameters of the sea ice layer, as shown in Figure 9.

In addition, we compared the crystal structures of natural and laboratory sea ice under orthogonally polarized light. Limited to experimental conditions, only one year of ice samples collected in Bohai Bay was compared. The comparison results are shown in Figure 10.

From the comparison of Figure 10a,b, the crystals parallel to the sea ice growth direction are all free of interruptions and overlaps and are of similar lengths. Figure 10b,c,e,f are essentially identical in ice crystal morphology and grain size. This indicates that the temperature conditions, freezing rate, and hydrodynamic conditions of the laboratory ice samples are basically the same as those of the natural ice samples.

The average elastic modulus, density, and Poisson’s ratio for each type are summarized in Table 2, showing clear distinctions among ice types.

The laboratory-prepared ice was classified into multi-year sea ice, peak sea ice, one-year sea ice, and decayed sea ice based on temperature and salinity thresholds. Comparative analysis showed that the densities of laboratory-prepared multiyear ice (0.885 ± 0.055 g/cm^3^) aligned with the field observations by Kern et al. [30]. Similarly, the elastic modulus and Poisson’s ratio measurements were consistent with previous studies under similar temperature conditions [31,32]. The ice crystal structure within one year of ice is similar to that of measured sea ice in the Bohai Sea.

### 3.2. Acoustic Properties of Sea Ice Layers

Accurate measurement of acoustic properties, particularly shear wave velocity ($V_{s}$) and longitudinal wave velocity ($V_{l}$), is vital for modeling sea ice behaviors. Prior to stratified ice sample measurements, a standard glass sample measurement test is required to verify the accuracy of the measurement instrument. The standard glass sample device is shown in Figure 11a, and the transducer fixed position is shown in Figure 11b.

Standard Glass Sample Experiment:

A custom-developed portable acoustic testing device was calibrated using a standard glass sample (known velocities: $V_{s}$ = 1500 m/s, $V_{l}$ = 3100 m/s). Measurement errors were controlled within ±12 m/s by averaging multiple readings, ensuring high precision for ice layer testing.

Measurement results are shown in Table 3:

The mean values of shear and longitudinal sound velocity measurements of the ten groups of glass samples were 1492.75 m/s and 3087.073 m/s, respectively. The standard deviations were less than 15 m/s, indicating that the error ranges were less than 1%. These results meet the requirements for high-precision experiments.

2.Acoustic Behavior of Simulated Sea Ice Layers:

Tests were conducted across frequencies (25–100 kHz) and temperatures (−6 °C and −14 °C). Ice sample measurements are shown in Figure 12a–d.

Based on the analysis of the measured information, the speed of sound is significantly higher for multi-year ice. More dominant, however, is the effect of temperature, which decreases $V_{l}$ by 946.5 m/s and $V_{s}$ by 263.4 m/s as the temperature increases. Meanwhile, the transverse and longitudinal speeds of sound differ greatly in the same medium [33].

Frequency has no significant effect on $V_{l}$ but shows some positive correlation for $V_{s}$, probably due to the porous ice structure or the modal effect of the sensor.

Measurements of the speed of sound in different ice samples also demonstrated the acoustic parameter measurement capabilities of the portable Acoustic Parameter Measurement Equipment.

### 3.3. Deep Learning Simulation of Acoustic Parameters

In order to improve the accuracy of velocity inversion and to recognize subtle parameter variations, we used ANN, ResNet, and SE-ResNet for training, respectively. Figure 13 shows the training loss and validation loss for different models.

Evaluation of ANN, ResNet, and SE-ResNet model loss results using a training-independent test set, for $V_{l}$ are 0.0197, 0.0132, and 0.0119. The improvements over ANN were 32.99% and 39.59%. For $V_{s}$, the average test set losses were 0.0575, 0.0480, and 0.0432. The improvements over ANN were 16.52% and 24.87%.

ANNs (Figure 13a,b) have a simpler structure, and the training loss decreases faster, but the gradient starts to disappear after a long time of training, and the loss cannot be reduced further. The ResNet and SE-ResNet models (Figure 13c–f) are slightly less stable at the beginning of training, and there is room for further decrease in validation loss compared to the ANN after multiple rounds of training

As a result, the SE-ResNet model has excellent stability and prediction accuracy over more rounds of training. This highlights the potential of deep learning models to capture complex nonlinear relationships in ice acoustic data, which can be applied to the inversion of in-ice acoustic parameters.

## 4. Discussion

In this paper, to overcome the difficulties of in situ measurements of sea ice, a solution is proposed in three dimensions. The results validate the effectiveness of the proposed approaches while highlighting areas for further refinement.

Laboratory Ice Sample Preparation

The laboratory-prepared sea ice samples demonstrated physical properties that align closely with field measurements under certain conditions. For instance, Poisson’s ratio is consistent with in situ observations of polar sea ice; density and elastic modulus of multi-year sea ice samples are consistent with field measurements in the range of −5 °C to −10 °C.

However, discrepancies were observed for one-year sea ice density and elastic modulus in the −10 °C to −15 °C range, which were lower than field-reported values. These differences may stem from two factors:

The first is the inadequate control of residual salinity within the ice during its formation. The second is the lack of controlled addition of impurities typical of polar sea ice (e.g., organic matter or sediment).

Future research should focus on refining the preparation of sea ice samples by simulating the salinity and impurity composition of polar seawater more accurately. This step is crucial for producing laboratory sea ice samples that better replicate natural conditions.

Acoustic Parameter Measurements

The portable acoustic measurement system developed in this study proved effective for determining the acoustic properties of sea ice layers.

The system, based on intrinsic radial vibration modes of transducers and a time-difference-of-arrival algorithm, achieved an error margin of less than 1% (mean error < 15 m/s) when validated with standard glass samples. At the same time, measurements of the laboratory sea ice layer show that temperature significantly affects $V_{s}$ and $V_{l}$ decreased as temperature rose, with multi-year sea ice exhibiting higher velocities than decaying sea ice. The effect on frequency suggests that there is more likelihood to be a frequency dependence for $V_{s}$ than for $V_{l}$.

The results for in-ice acoustic characterization also demonstrate the reliability of the system and its potential for application in field studies.

Acoustic Parameter Inversion with SE-ResNet

To enhance the inversion and prediction of acoustic parameters, ResNet and SE Attention were employed and compared with ANN. Among them, ResNet and SE-ResNet have a significant improvement in accuracy and stability:

Independent dataset evaluations show that the $V_{l}$ average test loss for ResNet and SE-ResNet model training is reduced by 0.0065 and 0.0078, $V_{s}$ loss reduced by 0.0095 and 0.0143, respectively, compared to the ANN model, which suggests that the modified models are more capable of generalization.

The SE-ResNet model outperforms the conventional ANN, especially on small datasets, thanks to its deeper network structure and better feature weight extraction mechanism, which shows its robustness and applicability to handle complex nonlinear relationships in ice sound data.

Implications and Future Directions

This study demonstrates the feasibility of using laboratory-prepared ice samples and advanced machine learning techniques to investigate and predict the acoustic properties of sea ice. The integration of ResNet models provides a powerful tool for handling small datasets and capturing intricate parameter relationships, offering significant potential for future research. However, refinements in both the physical ice preparation process and the ResNet architecture are needed to achieve greater accuracy and generalizability.

Moving forward, efforts should focus on:Improving the simulation of seawater conditions during ice preparation;Incorporating field-validated corrections for the ResNet model to optimize generalization and computational efficiency;Expanding the dataset size to enhance model robustness and predictive power.

Overall, these findings help to reduce the difficulty of measuring sea ice acoustic parameters and contribute to the monitoring of sea ice acoustic properties.

## 5. Conclusions

This study addresses the challenges posed by in situ measurements of sea ice layers, including the high logistical demands and harsh environmental conditions, by proposing innovative solutions across three aspects: sea ice layer simulation, measurement, and inversion.

First, a laboratory-based method for fabricating sea ice samples under simulated polar conditions was developed. This method replicates the unidirectional heat exchange process that is characteristic of natural sea ice formation and incorporates brine rejection by adjusting the salinity of the ice-making solution. The fabricated ice layers exhibit properties consistent with field measurements, including Poisson’s ratio, the density of multi-year sea ice, and the Elastic modulus at higher temperature ranges.

Second, a portable device for measuring the acoustic propagation parameters of sea ice layers was designed. Utilizing a modular design and a custom-made radial eigenmode vibration transducer, the device achieves high adaptability to extreme polar conditions and ensures a measurement accuracy exceeding 1%.

Finally, the SE-ResNet model was introduced. By integrating residual modules into the traditional artificial neural network architecture, the proposed method enhances both the accuracy and stability of parameter inversion. Compared to conventional models, the shear and longitudinal wave models exhibit accuracy improvements of 24.87% and 39.59%, respectively. This advancement reduces the workload of polar field measurements and provides a more efficient computational framework for sea ice parameter inversion.

## Figures and Tables

**Figure 1 sensors-25-05663-f001:**
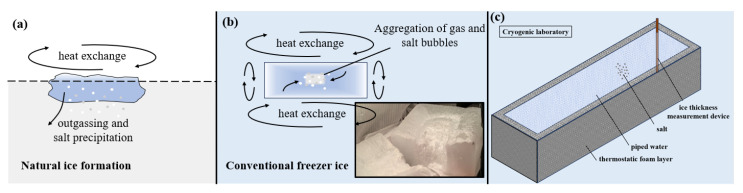
Natural icing and traditional ice-making methods. (**a**) Natural icing. (**b**) Traditional ice production. (**c**) Improved ice-making containers.

**Figure 2 sensors-25-05663-f002:**
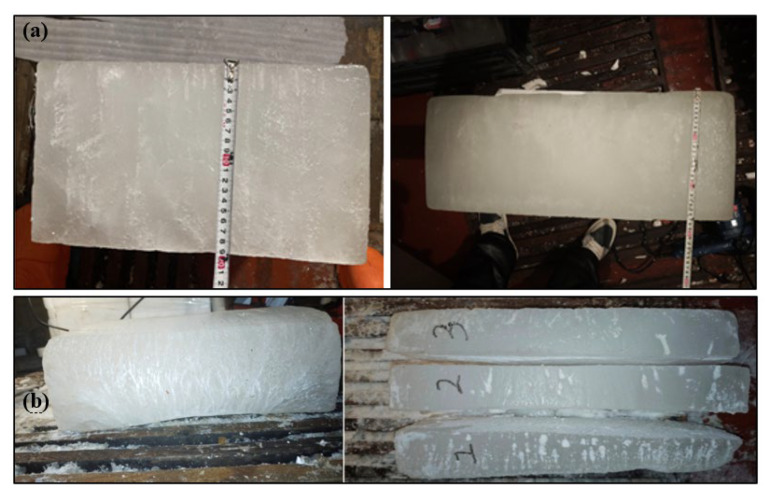
Preparation and cutting of sea ice layers. (**a**) Thickness measurement of sea ice layers prepared with ICE1 and ICE2 solutions. (**b**) Results of sea ice layers segmentation.

**Figure 3 sensors-25-05663-f003:**
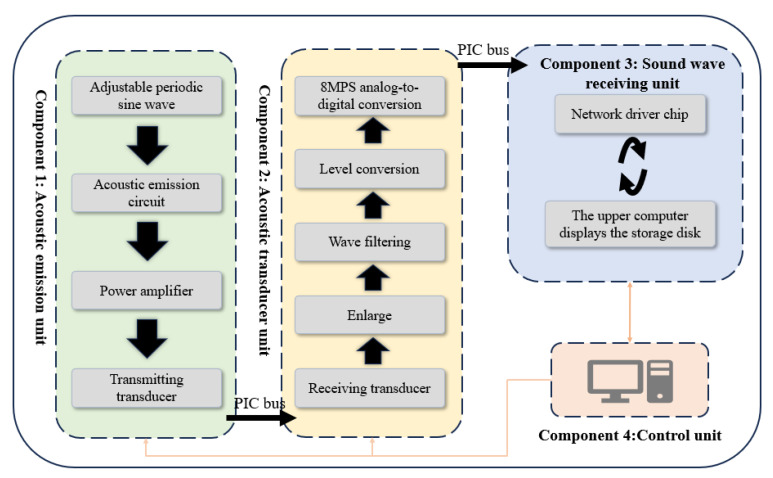
Acoustic parameter measurement system components.

**Figure 4 sensors-25-05663-f004:**
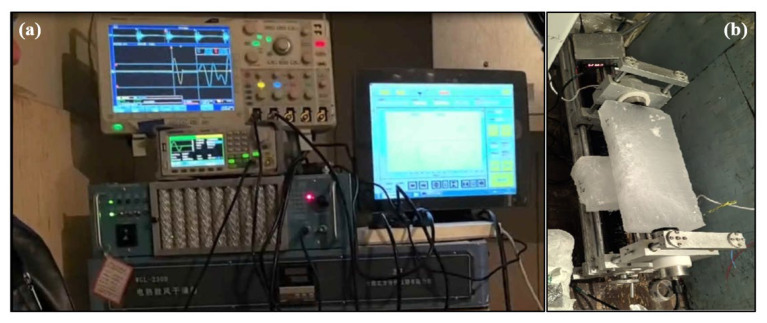
Diagram of the acoustic parameter measurement equipment. (**a**) Shows the constructed measurement equipment. (**b**) Shows the ice layer fixed to the measurement scale with transmitting and receiving transducers fixed at each end.

**Figure 5 sensors-25-05663-f005:**
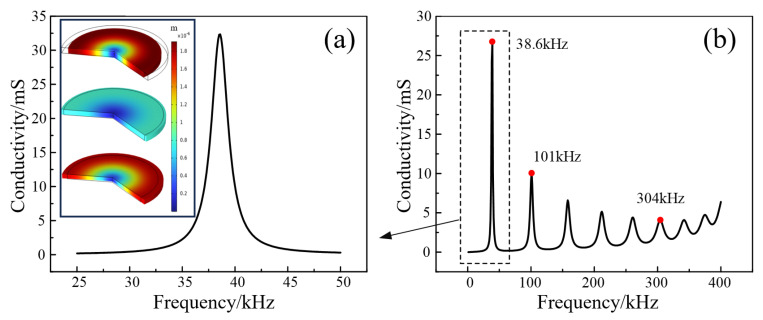
Conductance plot of a customized transducer. (**a**) Conductance plot of the 25–50 kHz transducer, where the vibration mode is mainly radial at 28.6 kHz. The thumbnail represents the amplitude of the transducer's radial vibration. (**b**) Conductance plot of the 0–400 kHz transducer.

**Figure 6 sensors-25-05663-f006:**
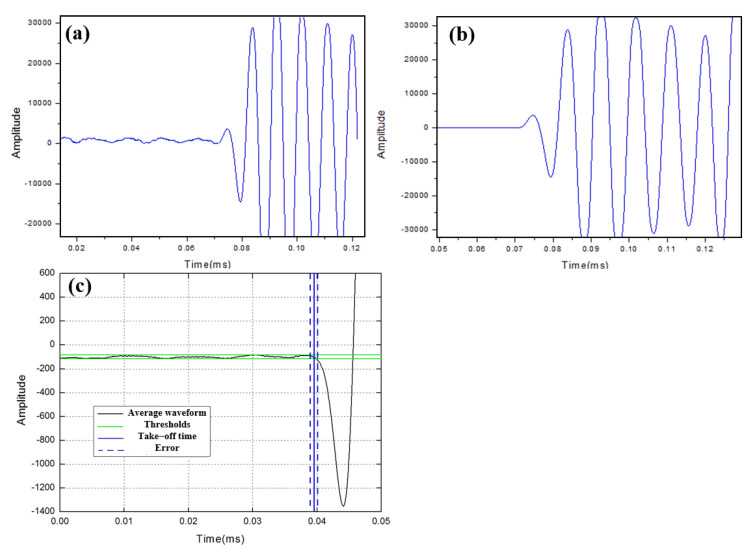
Repeated measurement take-off time for 20 cm × 10 cm × 10 cm samples for estimating error. (**a**) Single measurement results. (**b**) Repeated measurements averaged results. (**c**) Mean result fluctuation range and measurement error.

**Figure 7 sensors-25-05663-f007:**
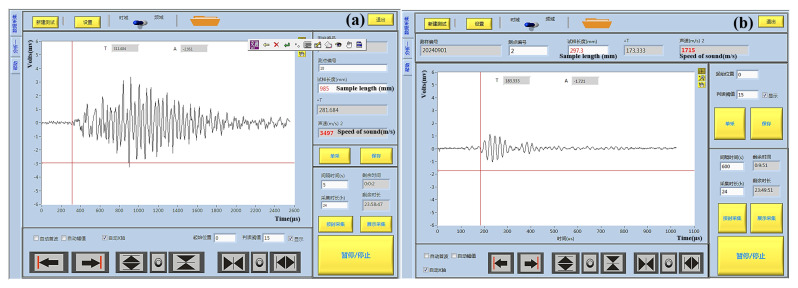
Measurements of the speed of sound with the integrated measurement program. (**a**) Measurement of the longitudinal sound velocity at an angle of incidence of 0° using a 25 kHz source. (**b**) Measurement of shear sound velocity at an angle of incidence of 180° using a 44 kHz source.

**Figure 8 sensors-25-05663-f008:**
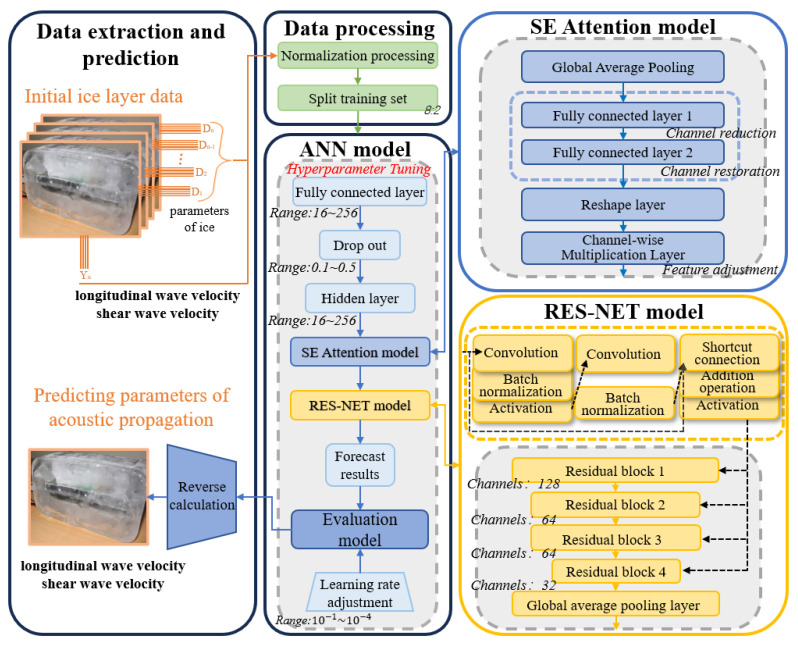
ResNet model based on sea ice physical parameter prediction.

**Figure 9 sensors-25-05663-f009:**
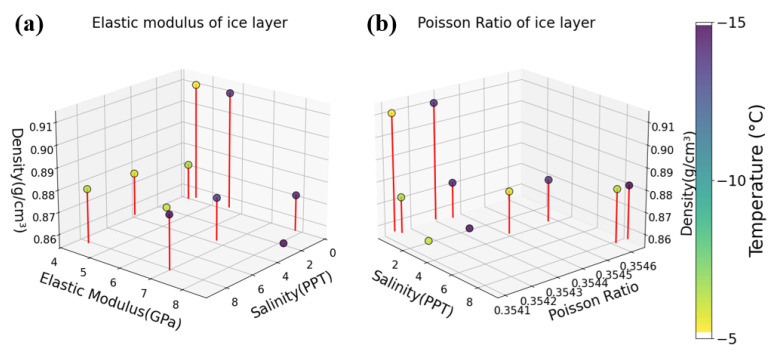
Distribution of physical parameters underlying the ice sheet (**a**) Ice sheet salinity, elastic modulus, density-temperature relationship. (**b**) Ice sheet salinity, Poisson’s ratio, density, and temperature relationships.

**Figure 10 sensors-25-05663-f010:**
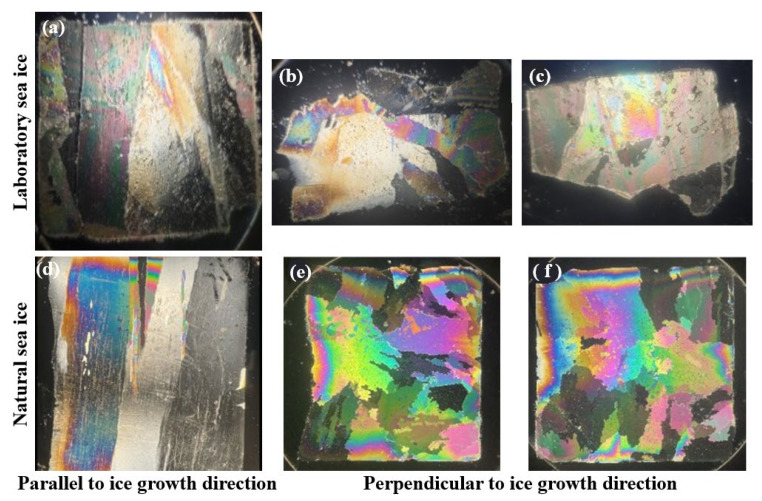
Comparison of laboratory sea ice and natural sea ice crystal structures. (**a**) is the crystal structure of laboratory sea ice parallel to the ice growth direction. (**b**,**c**) is the crystal structure of laboratory sea ice perpendicular to the direction of ice growth. (**d**) is the natural sea ice crystal structure parallel to the ice growth direction. (**e**,**f**) is the natural sea ice crystal structure perpendicular to the ice growth direction.

**Figure 11 sensors-25-05663-f011:**
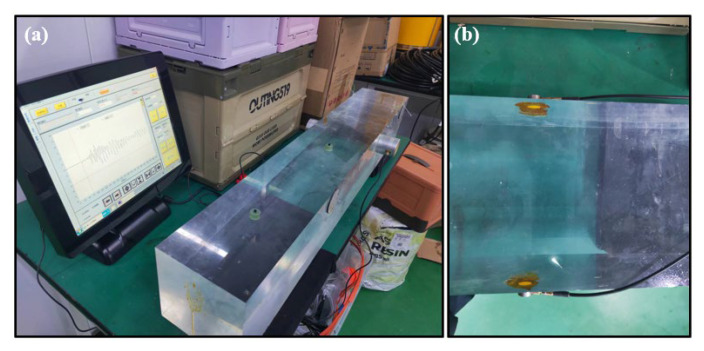
Standard glass sample measurement pattern (**a**) Overview of experimental equipment for glass ice samples. (**b**) Fixed position of transmitter-receiver transducer.

**Figure 12 sensors-25-05663-f012:**
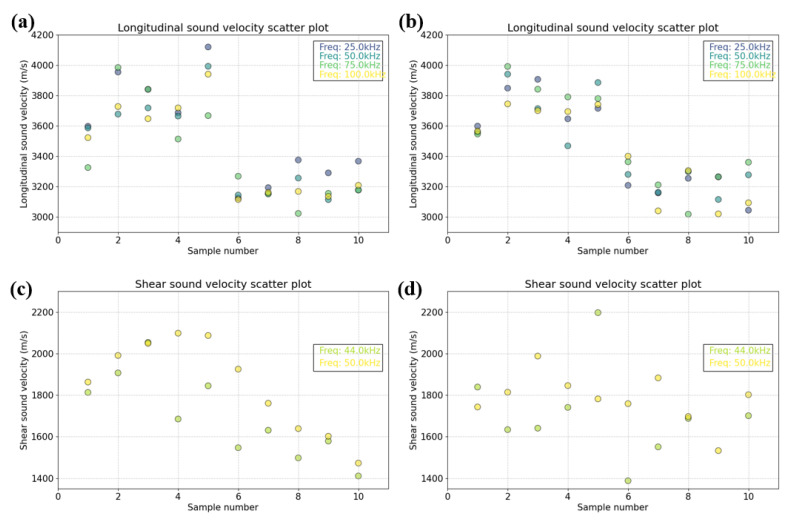
Samples of acoustic parameters in ice, with the horizontal axis indicating the different ice classified according to Table 2 and the vertical axis indicating the acoustic parameter. (**a**) Measured longitudinal sound velocity of sound sources of different frequencies in ice at an emission angle of 0 degrees. (**b**) Measured longitudinal sound velocity of sound sources of different frequencies in ice at an emission angle of 180 degrees. (**c**) Measured shear sound velocity in ice for sources of different frequencies at an emission angle of 0 degrees. (**d**) Measured shear sound velocity in ice at 180 degrees for different frequency sources.

**Figure 13 sensors-25-05663-f013:**
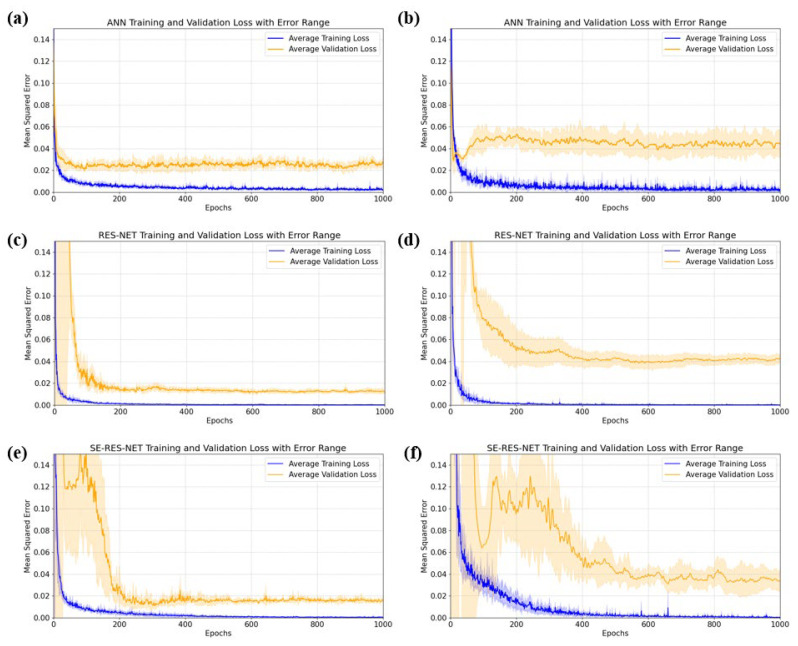
Training loss and validation loss, where orange represents the average validation loss over five training sessions, blue represents the average training loss ofiveer 5 training sessions, and light orange and light blue represent the validation loss error and the training loss error, respectively. (**a**) Training results of ANN for $V_{l}$. (**b**) Training results of ANN for $V_{s}$. (**c**) Training results of ResNet for $V_{l}$. (**d**) Training results of ResNet for $V_{s}$. (**e**) Training results of SE-ResNet for $V_{l}$. (**f**) Training results of SE-ResNet for $V_{s}$.

**Table 1 sensors-25-05663-t001:** Simulated sea ice classification table.

Simulated Sea Ice Type	Ambient Temperature	Preparation Salinity
Decaying Sea Ice	−5~−10 °C	0~5 ppt
One-Year Sea Ice	−5~−10 °C	5~10 ppt
Multi-Year Sea Ice and Peak Sea Ice	−10~−15 °C	0~5 ppt
One-Year Sea Ice	−10~−15 °C	5~10 ppt

**Table 2 sensors-25-05663-t002:** Ice stratification results.

Sample	No.	Layer	Salinity (ppt)	Temperature (°C)	Density (g/cm^3^)	Elastic Modulus (GPa)	Poisson’s Ratio	Ice Type
ICE1	1	Top	0.5	−13.9	0.911	5.8105	0.35427	Multi-year and Peak
	2	Mid	0.9	−14.2	0.874	8.1215	0.35432	Multi-year and Peak
	3	Bottom	2.7	−14.8	0.858	8.4222	0.3543	Multi-year and Peak
ICE2	4	Top	4.5	−13.6	0.877	7.0054	0.35453	Multi-year and Peak
	5	Mid	8.9	−14.9	0.882	7.3370	0.35462	One-year
ICE1	6	Top	0.5	−5.2	0.911	4.6742	0.35411	Decaying
	7	Mid	0.9	−6.4	0.874	4.5752	0.35412	Decaying
	8	Bottom	2.7	−6	0.858	4.5775	0.35414	Decaying
ICE2	9	Top	4.5	−5.7	0.877	4.2587	0.35437	Decaying
	10	Mid	8.9	−6.2	0.882	4.6700	0.35457	One-year

**Table 3 sensors-25-05663-t003:** Measurement results of standard glass samples.

No.	Measurement of Shear Sound Velocity	Measurement of Longitudinal Sound Velocity	No.	Measurement of Shear Sound Velocity	Measurement of Longitudinal Sound Velocity
1	1482.6 m/s	3090.5 m/s	6	1488.4 m/s	3085.8 m/s
2	1493.4 m/s	3081.2 m/s	7	1494.5 m/s	3085.9 m/s
3	1485.2 m/s	3100.1 m/s	8	1493.9 m/s	3085.1 m/s
4	1501.3 m/s	3078.6 m/s	9	1498.2 m/s	3085.11 m/s
5	1496 m/s	3093.3 m/s	10	1494 m/s	3085.12 m/s

## Data Availability

Relevant data can be obtained by contacting the author’s e-mail address.

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
