# Peer review of "Inversion of Shear and Longitudinal Acoustic Wave Propagation Parameters in Sea Ice Using SE-ResNet"

_sensors, 2025, doi:10.3390/s25185663_

Round 1
Reviewer 1 Report (Previous Reviewer 1)
Comments and Suggestions for Authors I have read the authors' responses. There were two points in the comments:
1- the samples were too small: The authors clarified that the samples measured 20cm x 10cm x 10cm. The wavelength at 25 kHz at the given speed speed is 12 cm. The sample size needs to be bigger than a few wavelengths in all dimensions. Therefore, the comment stands.
2- the Young's modulus values are too low: The authors have updated Table 2 with more reasonable values.
As shown above, the authors were able to satisfy one of the two comments. My recommendation for rejection is unchanged.
Author Response
Please see attachment

Reviewer 2 Report (New Reviewer)
Comments and Suggestions for Authors
The article is devoted to the development of a method for measuring the mechanical properties of ice using ANN, which is based on the analysis of the speed of sound waves in ice samples. The work makes a positive impression. The introduction contains a sufficient amount of information to understand the relevance and current state of the problem under study. Descriptions of the methods and experimental approaches used are presented quite clearly, including a procedure for preparing samples for experiments. There is a comparison with calibration sample and results of other authors. The bibliography mainly consists of modern publications.
At the same time, it remains not entirely clear why sample #5 in Figure 11a and sample #2 in Figure 11b have the highest longitudinal sound speed. According to the standard formula from the theory of elasticity, based on the data in Table 2, the maximum speed should be in sample #3, which has the highest elastic modulus. This point requires some clarification from the authors.
A small technical inaccuracy: in the legends to Figure 11 the values are given in Hz, whereas according to the context they should be kHz.
Author Response
Please see attachment

This manuscript is a resubmission of an earlier submission. The following is a list of the peer review reports and author responses from that submission.
Round 1
Reviewer 1 Report
Comments and Suggestions for Authors
The application of ANN and RESNET seem quite straightforward. The problem lies in the acoustic measurements.
The ice samples are too small for the frequencies used. Looking at Fig. 5, the travel time across the sample is about 0.03 ms. At a speed of 3000 m/s, this makes the path length through the sample about 9 cm. The wavelength at 25 kHz and the same speed is 12 cm. Therefore the path length is less than a wavelength. It is not possible to reliably measure wave speed over such a small path length. At the highest frequency, 100 kHz, the wavelength is 3 cm, and the path is still only 3 wavelengths. The situation is better for shear waves, which has roughly half the speed of the compressional wave. Another problem is the width of the sample, as shown in Fig. 4b, which is smaller than the path length by a factor of about 10. Therefore, there are serious doubts that the wave speeds measured are not the true values. There is no problem with the calibration glass block because in comparison to the ice samples, it is quite large.
The elastic modulus values in Table 2 appear to be too low compared to the measured speeds in Fig. 9. This needs some explanation. For example:
line 1. Elastic modulus 307E6, poissons ratio 0.354, sound speed should be
sqrt((307e6*(1-0.354)/(2*(1+0.354)*(0.5-0.354)))/900)
=746 m/s
This is clearly too low.
Reviewer 2 Report
Comments and Suggestions for Authors
This ms developed an improved sea-ice preparation container to control heat exchange, and proposed an innovative approach using laboratory-prepared sea-ice samples, acoustic measurement techniques, and a deep learning model (RES-NET) for the simulation and inversion of shear wave propagation parameters in polar sea-ice. Specific comments and suggestions are as follows.
1. What is the basis for classifying simulated sea ice? How to determine what simulated conditions correspond to these decaying, one-year, and multi-year phases classifications?
2. The structural position of section 2.3 needs to be adjusted. The section 2.3 has obtained the conclusions of RES-NET before describing the RES-NET model.
3. Figure 6 is poorly readable and the model structure diagram could be improved.
4. The article is loosely structured and poorly readable. For example, the presentation in Section 2.3.2 could be more concise. The RES-NET is a commonly used classical model and does not require much description of the basic tuning parameter content. The small amount of original content is listed in points, which is loosely structured and poorly readable.
5. The article uses the production of sea ice samples under simulated polar conditions as input data for training the RES-NET model, what is the difference between this and training with other conventional models/observations? What are the implications?
6. Why did the authors use the RES-NET model instead of other deep learning models? Is there any particular significance?
Comments on the Quality of English LanguageThe presentation of the article could be improved, the structure needs further refinement and is currently too loosely expressed and less readable.